# PROJECTIVE SYMBOLIC REGRESSION:
# SOLVING HIGH-DIMENSIONAL PDES BY LEARNING FROM LOW-DIMENSIONAL PROJECTIONS

## ABSTRACT

Symbolic regression (SR) provides a powerful means for uncovering the underlying mathematical structure of physical systems, such as those governed by partial differential equations (PDEs). However, applying SR directly to high-dimensional PDEs remains intractable due to the curse of dimensionality. To address this, we propose Projective Symbolic Regression (PSR), a novel framework that solves high-dimensional PDEs by learning from low-dimensional projections. PSR first generates multiple projections of the PDE solution data by fixing subsets of input variables. Symbolic regression is then applied to each projection to extract compact, localized functional components. These components are subsequently composed into a unified global expression through a higher-level symbolic program. Critically, the final composition is constrained by minimizing the PDE residual error, ensuring physical validity. Empirical results demonstrate that PSR not only improves predictive accuracy over conventional methods but also yields interpretable models that reveal the compositional structure of the underlying physical dynamics.

## 1 INTRODUCTION

Partial Differential Equations (PDEs) are fundamental tools to describe a wide range of natural phenomena and engineering problems, such as how heat transfers in a cooking pot (Hein et al., 2015) or how pressure acts on the surface of an aircraft during flight (Karkoulias et al., 2023). Obtaining analytical solutions to PDEs is of great significance for scientific understanding, as they offer explicit insights into how the physical field governed by PDE varies over time and space. Such insights provide a clear foundation for engineers and scientists to grasp the underlying physical principles and make more informed and effective design decisions (Ganie et al., 2024).

Deriving analytical solutions for real-world PDEs poses significant challenges, as the inferring process requires advanced mathematical expertise and sophisticated analytical techniques (Roach, 1982; Jena, 2025; Wong, 2022). More importantly, for a large class of PDEs, no analytical solution is known to exist within the current theoretical frameworks. Therefore, finding more efficient approximate methods (Temam, 2024; Gelbrecht et al., 2021) for solving PDEs continues to be a central goal in scientific and engineering research.

While traditional numerical solvers (Dhatt et al., 2012; Bathe, 2007; Jagota et al., 2013) and deep learning-based methods (Cuomo et al., 2022; Cai et al., 2021; Raissi et al., 2019) are widely applied for solving PDEs, both paradigms share a fundamental limitation: they only provide approximate numerical solutions, which restricts interpretability and limits the theoretical insights that can be drawn from the results. Symbolic Regression (SR) emerges as a distinct paradigm to address this gap. By reframing the solution process as a search for a symbolic expression satisfying the governing constraints of PDE and observed dataset, SR can derive an explicit, human-readable approximate solution in the form of a symbolic expression (Oh et al., 2024; Tsoulos & Lagaris, 2006). The symbolic expression offers direct physical insight, bridging the interpretability gap inherent in traditional numerical and deep learning-based methods.

Despite its interpretability and success in low-dimensional problems, SR still faces challenges when applied to high-dimensional PDEs (Cao et al., 2024a). The core difficulty lies in the exponentially expanding search space of symbolic expressions, which grows rapidly with the number of

variables (Jiang & Xue, 2023; Udrescu et al., 2020). The existing works to mitigate this curse of dimensionality can be categorized into two main directions. On one hand, some recent SR methods proposed general decomposition strategies, such as divide-and-conquer or vertical discovery (Udrescu et al., 2020; Jiang & Xue, 2023). However, these methods fail to utilize the intrinsic physical structure inherent in PDEs, where variable dependencies are not arbitrary. The decomposition strategies for solving PDEs still leave room for improvement. On the other hand, more specialized approaches like HD-TLGP (Cao et al., 2024a) have shown success by using a known one-dimensional solution to guide the search in higher dimensions. While effective in certain cases, a key limitation of this approach is its reliance on a known low-dimensional PDE solution, which restricts its applicability to real-world physical systems where such priors are often unavailable or difficult to obtain with sufficient accuracy.

Therefore, in this work, we aim to reduce the difficulty of SR by proposing a new decomposition strategy grounded in the foundational principle of physical fields described by PDEs, without relying on low-dimensional prior solutions. The foundational principle supporting our approach is known as **sparsity-of-effects** (Montgomery, 2017; Blatman & Sudret, 2009), which means that the behavior of a complex physical system can often be decomposed into a combination of simpler effects that are dominant in lower-dimensional spaces. (Box et al., 2005; Saltelli et al., 2008; Soboĺ, 1993; Szabo & Ostlund, 1996) This principle has been extensively validated across a wide range of scientific domains. In engineering, it forms the basis of classical experimental design, where system behavior is assumed to be governed primarily by a limited number of main effects and low-order interactions (Box et al., 2005). In mathematics, it is strictly formalized through global sensitivity analysis, where the majority of output variance is typically attributed to first-order and second-order effects (Saltelli et al., 2008; Soboĺ, 1993). In theoretical physics, it is implicitly employed in methods such as the Hartree-Fock approximation (Lykos & Pratt, 1963), which constructs solutions by first capturing dominant mean-field contributions and then incorporating higher-order corrections (Szabo & Ostlund, 1996).

Motivated by the above cross-disciplinary insight, we propose **P**rojective **S**ymbolic **R**egression (PSR), a novel framework that puts the sparsity-of-effects principle into practice to address the curse of dimensionality. PSR begins by projecting the dataset from observations or numerical simulations corresponding to the high-dimensional PDE onto lower-dimensional subspaces. We simplify the problem for each projection by treating all variables outside of that subspace as fixed parameters. This allows us to apply a local symbolic regression that focuses only on the active variables. The goal of this local search is to discover a functional component, a simpler mathematical expression that captures how these active variables influence the PDE's solution within that specific projection. With the low-dimensional components identified, PSR performs a final global symbolic regression to discover the optimal way to compose them into a high-dimensional expression. This search is constrained by the governing PDE to ensure the resulting expression is a physically valid global solution. The main contributions of our work can be summarized as follows:

• We propose Projective Symbolic Regression (PSR), a novel framework designed to mitigate the curse of dimensionality in solving high-dimensional PDEs. PSR reframes the problem by decomposing the high-dimensional search space into a series of lower-dimensional projections, making the discovery of symbolic solutions more tractable.

• We design a hierarchical discovery process that operates in two stages. First, a local symbolic regression is applied within each projection to identify low-dimensional functional components. Second, a global symbolic regression discovers the optimal composition of these components, guided by the governing PDE to ensure a physically valid solution.

• We demonstrate through experiments that the discovered symbolic solutions offer deep physical insight by effectively revealing how the physical field governed by the PDE interacts across its various dimensions.

## 2 PRELIMINARIES

This preliminary study is structured in three parts: defining the Partial Differential Equation problem, outlining the current landscape of Symbolic Regression methods, and reviewing related work for solving PDEs to establish the background for our approach.

## 2.1 Partial Differential Equation

A partial differential equation (PDE) is a mathematical tool used to describe physical phenomena. Let $\mathcal{N}$ be a partial differential operator. A PDE can be formally expressed as:

$$\mathcal{N}(u(\mathbf{x})) = 0, \ \mathbf{x} = (x_1, ..., x_d, t) \tag{1}$$

where $u$ is the unknown function to be solved and $\mathbf{x}$ is a vector of independent variables, often including spatial coordinates and time. Real-world physical systems are typically modeled in up to three spatial dimensions. Problems involving two or more spatial dimensions ($d \geq 2$) are commonly considered high-dimensional. Solving a PDE refers to finding a mathematical expression for $u(\mathbf{x})$ that satisfies the equation, allowing us to understand and predict how the physical system behaves over time and space.

## 2.2 Symbolic Regression

Several major approaches dominate the field of Symbolic Regression(SR). The most classical is Genetic Programming (GP), an evolutionary algorithm that evolves a population of expression trees that evolve tree-structured expressions through selection, crossover, and mutation (Zhang et al., 2022; Zhong et al., 2025). However, GP suffers from inefficient exploration and poor scalability, due to its reliance on random search and CPU-based execution. To address these limitations, neural network-based approaches have been introduced. Deep symbolic regression (DSR) (Petersen et al., 2019) leverages deep reinforcement learning to guide expression generation, enabling GPU acceleration and improved search efficiency. Hybrid frameworks combining GP and DSR aim to leverage the broad search ability of evolutionary algorithms and the targeted optimization strength of neural methods (Landajuela et al., 2022). More recently, pre-trained transformer models (Valipour et al., 2021; Kamienny et al., 2022) have been proposed to enable rapid inference by mapping numerical data to symbolic expressions in a single forward pass. Despite their differences, all these methods fundamentally struggle when the number of input variables becomes large, a problem known as the curse of dimensionality.

**Decomposition Strategies in Symbolic Regression.** Some recent symbolic regression methods have introduced decomposition strategies to mitigate the curse of dimensionality. Udrescu et al. (2020) uses a recursive divide-and-conquer approach. It first checks for simple patterns like symmetry or separability (such as $f(x, y) = g(x) + h(y)$ or $f(x, y) = g(x)h(y)$). If such patterns are found, the method splits the problem into smaller parts that can be solved independently. Jiang *et al.* follow a vertical discovery strategy. They start with reduced equations that use only a few variables, and then gradually add more variables one at a time (Jiang & Xue, 2023; 2024; Jiang et al., 2024). Although general strategies offer broad applicability, they fall short in capturing the intrinsic structure of PDEs, where variable dependencies are not arbitrary. This highlights the need for a decomposition strategy customized to the physics of PDEs for improved performance.

## 2.3 Related Work

**Traditional numerical methods** discretize a continuous problem domain into a finite mesh of points or cells, obtaining approximate solutions that satisfy the PDE at each mesh point. The mesh structure and resolution critically influence the accuracy, stability, and computational efficiency of the solution, as well as the types of PDEs that can be effectively addressed (Braun & Sambridge, 1995; Rezzolla, 2011). Techniques such as the Finite Element Method (FEM) (Bonito et al., 2024; Liu et al., 2022; Kudela & Matousek, 2022), Finite Difference Method (FDM) (Gedney, 2022; Meiliang et al., 2021; Hoang, 2025), and Finite Volume Method (FVM) (Cardiff & Demirdžić, 2021; Muhammad, 2021) have long served as the backbone of computational physics and engineering, offering robust and well-established frameworks for simulating complex physical systems across diverse domains.

**Deep learning-based methods** can directly learn the solution trajectories of PDE from observed data, often achieving orders-of-magnitude speedups compared to traditional numerical methods. Two mainstream neural network paradigms have been widely adopted for solving PDE: the family of PINN and Neural Operator.

Physics-Informed Neural Networks (PINNs) (Raissi et al., 2019) represent the PDE solution using a neural network whose loss functions explicitly incorporate the governing equations. By minimizing

both the residuals of the PDE and the errors in initial/boundary conditions or observational data, PINNs learn solutions that are physically consistent. The architecture of PINNs has evolved beyond standard fully connected networks to generative adversarial networks (Yang & Perdikaris, 2019), and recurrent neural networks Geneva & Zabaras (2020). Kharazmi *et al.* incorporated the variational form of PDE to reduce residual errors Kharazmi et al. (2019; 2021). Jagtap *et al.* designed adaptive activation functions to accelerate the minimization process of the loss values (Jagtap et al., 2020).

Neural Operators learns the mapping between function spaces, allowing for the rapid solution of entire families of PDEs (Li et al., 2020b; Kovachki et al., 2023). DeepONet is the first neural operator that combines two neural networks: a branch network that encodes input functions and a trunk network that represents output coordinates (Lu et al., 2021a). Inspired by Green's function method, Li et al. (2020a) proposed Fourier neural operator (FNO), which parameterizes the convolution kernel of the integral operator directly in Fourier space and achieves superior accuracy compared to previous learning-based solvers. Cao et al. (2024b) proposed Laplace neural operator (LNO), which leverages the Laplace transform to decompose the input space and demonstrates improved performance over FNO.

**Symbolic regression-based methods** solve PDEs by discovering explicit mathematical expressions that both fit the observation dataset and PDE constraints, thereby offering interpretability that is often lacking in traditional numerical and deep learning-based methods. Oh et al. (2024) improved genetic programming to recover true analytic solutions of differential equations. Cao et al. (2023) built on the work of Oh et al. (2024) and added a pruning operator to avoid redundancy and increase diversity. Wei et al. (2024) proposed a novel reinforcement learning (RL)-based method for deriving closed-form symbolic solutions to differential equation. Cao et al. (2025) proposed NetGP, a hybrid framework that integrates deep symbolic regression (DSR) with genetic programming (GP) to solve PDE. However, these methods face significant challenges when applied to high-dimensional PDEs, primarily due to the vast and complex search space. To mitigate this issue, Cao et al. (2024a) proposed a transfer learning mechanism that transfers the structure of a one-dimensional analytical solution to guide the search for a high-dimensional PDE solution. While effective in certain cases, this approach is inherently limited, as most real-world physical systems governed by PDEs lack a known single-dimensional PDE. Therefore, in this work, we focus on reducing the difficulty of SR solely based on the available dataset and the form of high-dimensional PDE, without relying on low-dimensional prior solutions.

## 3 PROPOSED METHOD

In this work, we propose an incremental builder called Projective Symbolic Regression (PSR) to tackle the challenge of discovering a symbolic solution for a high-dimensional PDE, where the spatial dimension $d \geq 2$. The PSR is both data-driven and physics-informed. It leverages a dataset $D = (\mathbf{x}_i, u_i)$, which describes the behavior of the physical field and may be sourced from experimental measurements or numerical simulations. We design a decomposition strategy, grounded in the sparsity-of-effects principle, that implements a divide-and-conquer process through projection and hierarchical symbolic regression. The known partial differential operator $\mathcal{N}$ is then utilized as a key component of the evaluation function, guiding this search towards a physically consistent expression and making the entire framework physics-informed.

### 3.1 FRAMEWORK OVERVIEW

The framework of the proposed method (PSR) is illustrated in Figure 1. PSR first decomposes the high-dimensional problem by generating multiple low-dimensional projected datasets based on a set of predefined strategies (Stage 1). Based on these projected datasets, a Local Symbolic Regression engine generates a set of symbolic expressions for each projection. These expressions form the local components (Stage 2). Then, PSR treats these local components as high-level tokens and uses a Global Symbolic Regression engine to combine these tokens into full expressions. These expressions are substituted into the original PDE to evaluate how well they work, using a loss function that captures both data fit and PDE consistency. The evaluation results are then used to guide the global search towards a physically consistent solution (Stage 3). The whole process of PSR is shown in Algorithm 1. We now describe each of these stages in detail.

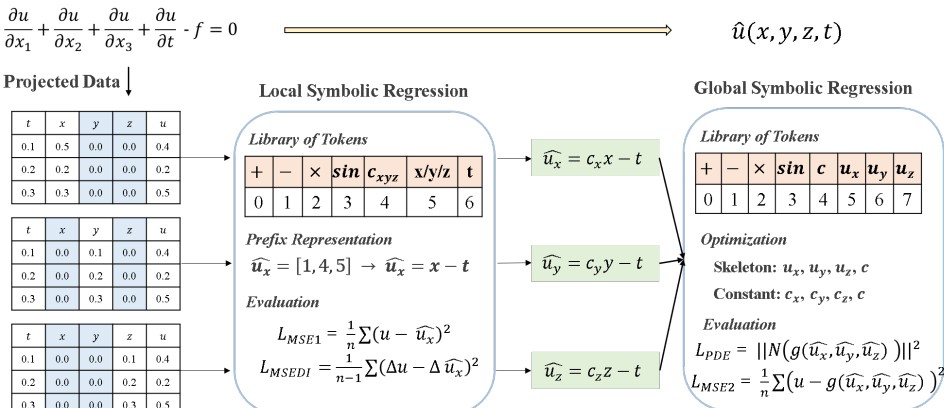

Figure 1: The hierarchical workflow of Projective Symbolic Regression (PSR). (Left) High-dimensional data is decomposed into multiple low-dimensional projected datasets. (Center) A Local Symbolic Regression engine operates on each projection, searching for the best-fitting expression (e.g., $\hat{u}_x$) using a library of primitive tokens and a prefix representation. (Right) The resulting local expressions are raised to become new tokens in a higher-level library for a Global Symbolic Regression search. This final stage optimizes both the compositional structure (skeleton) and the constants to discover the complete, high-dimensional symbolic solution.

---

**Algorithm 1** Projective Symbolic Regression (PSR)

---

1: **Input**: $\mathcal{N}$(PDE), $D$(Dataset), $\mathcal{S}$(Projection strategies).
2: **Output**: $\hat{u}$ (Optimal symbolic expression solutions).
3: $\mathcal{U} \leftarrow \emptyset$
4: **for** each strategy $S_j$ in $\mathcal{S}$ **do**
5: $\quad D_j \leftarrow$ GenerateProjection$(D, S_j)$
6: $\quad u_j \leftarrow$ LocalSymbolicRegression$(D_j)$
7: $\quad \mathcal{U} \leftarrow \mathcal{U} \cup \{u_j\}$
8: **end for**
9: $\hat{u} \leftarrow$ GlobalSymbolicRegression$(D, \mathcal{U}, \mathcal{N})$
10: **return** $\hat{u}$

---

### 3.2 PROJECTION DATA GENERATION

This stage implements the "divide" part of our strategy, motivated by a widely observed property of physical systems: their complex global behavior is often an aggregation of simpler, low-order interactions. This empirical principle, known as the "sparsity-of-effects," is supported by global sensitivity analysis, which shows that most output variance is explained by main effects and second-order terms. Inspired by this, we design a projection-based decomposition. By holding selected variables defined by projection strategies as constant and slicing the data into lower-dimensional subsets, we aim to isolate dominant low-order components. This allows us first to explore and symbolically capture how the solution behaves within these simpler subspaces.

As shown in Algorithm 1, each Projection Strategy $S_j$ determines how the data is partitioned. It specifies which variables are to be fixed, thereby defining the active variables whose relationship we aim to discover in the current subspace. For instance, in a spatio-temporal problem, a usual strategy retains one spatial dimension and the temporal variable $t$ active, while fixing all other spatial dimensions.

To construct a projected dataset $D_j$ under a given strategy $S_j$, we first identify the fixed variables $\mathbf{x}_{fixed}$ and the active variables $\mathbf{x}_{active}$. We then select the constant values $\mathbf{c}_j$ for the fixed variables. This is achieved by grouping the entire dataset $D$ according to the unique values of $\mathbf{x}_{fixed}$ and choosing the most populated group. This ensures that symbolic regression is performed on the most representative subset of the data. The resulting projected dataset is formed as $D_j = \{(\mathbf{x}_{active}, u(\mathbf{x}_{active}, \mathbf{x}_{fixed} = \mathbf{c}_j))\}$.

## 3.3 LOCAL COMPONENT DISCOVERY

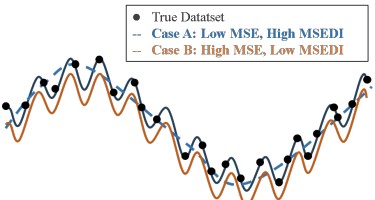

Figure 2: **Visual comparison of MSE and MSEDI for fitting a complex dataset.** The true dataset (black dots) exhibits a low-frequency trend with high-frequency oscillations. Case A (blue, dashed) captures only the coarse trend. Case B (red, solid) accurately captures the full dynamics. Despite a slightly higher MSE due to a minor offset, its near-zero MSEDI reflects a more physically faithful solution.

While our framework is compatible with any symbolic regression (SR) algorithm, in this work we employ a Neural-Guided Genetic Programming (NetGP) Cao et al. (2025). NetGP improves traditional genetic programming by integrating deep reinforcement learning that predicts promising symbolic operators and structures, thereby accelerating the evolutionary search and enhancing convergence toward optimal solutions.

For each projected dataset $D_j$, NetGP searches for a low-dimensional function $u_j(\mathbf{x}_{active}, \mathbf{x}_{fixed} = \mathbf{c}_j))$ that best fits the data. The fitness of each candidate expression is assessed using a composite loss comprising two terms. We first employ the Mean Squared Error (MSE) to quantify the difference between the predicted values and the true values (see Eq. 2). Since the full high-dimensional PDE residual cannot be computed on a low-dimensional slice, we introduce the Mean Squared Error of the first-order Difference (MSEDI), inspired by Fong et al. (2022). MSEDI evaluates the variance between the discrete derivatives of the true data and those of the candidate expression (see Eq. 3).

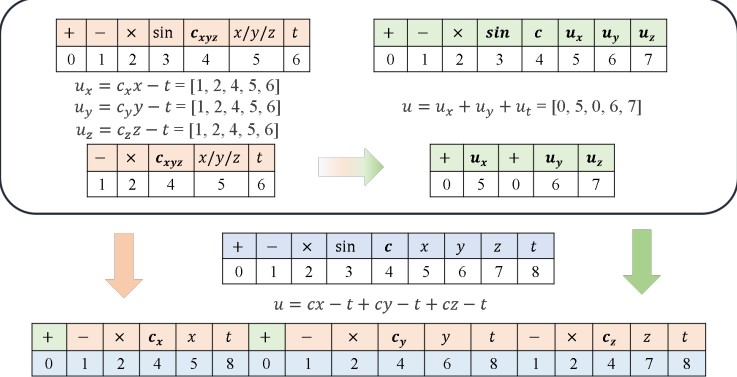

Figure 3: The substitution process for evaluating a global candidate solution. The Global SR engine discovers a compact expression using the local functions $(u_x, u_y, u_z)$ as high-level tokens. For evaluation, this expression is unfolded by substituting each high-level token with its corresponding full symbolic formula discovered in the local SR stage. This results in a complete, low-level expression that can be directly evaluated against the original high-dimensional dataset and the PDE operator.

$$\mathcal{L}_{\text{MSE1}} = \frac{1}{n} \sum_{i=1}^{n} \left( u_i - \hat{u}_{x,i} \right)^2 \tag{2}$$

$$\mathcal{L}_{\text{MSEDI}} = \frac{1}{n-1} \sum_{i=1}^{n-1} \left( \Delta u_i - \Delta \hat{u}_{x,i} \right)^2 \tag{3}$$

As shown in Figure 2, incorporating MSEDI provides a crucial advantage: it makes the fitness function sensitive to the dynamic patterns and local structure of the data, rather than solely to its

values. This aligns with the nature of PDEs, which impose constraints on derivatives. Thus, MSEDI can help guide the search towards solutions that are not only numerically accurate on the slice but also exhibit correct local behavior, thereby making them more appropriate for global composition.

As shown in Algorithm 1, this stage yields a set of functional components, denoted as $U = \{u_1, ..., u_j, ...\}$, where each $u_j$ characterizes the behavior of an unknown function within a specific low-dimensional subspace.

### 3.4 GLOBAL EXPRESSION COMPOSITION

This stage implements the "conquer" part of our framework, where the previously identified local components are integrated into a unified global solution. The set of local functions $\{u_j\}$, together with standard mathematical operators (e.g., $+, \times$), defines a new high-level token library. A global symbolic regression explores this new space to identify a compositional expression $g$ that combines these tokens, yielding the final solution $u(\mathbf{x}) = g(u_1, .., u_j, ...)$.

The evaluation of each candidate composition $g$ is performed in the original high-dimensional space. As illustrated in Figure 3, this involves a substitution process where each high-level token $u_j$ in $g$ is replaced by its full symbolic expression. This unfolds the compact high-level expression into a complete candidate solution u($\mathbf{x}$). After this unfolding, any free constants within u($\mathbf{x}$) are optimized to best fit the original dataset $D$.

The fitness of a candidate composition is evaluated using a composite loss function that integrates physical constraints and data accuracy. The first term, $\mathcal{L}_{\text{PDE}}$, quantifies the residual of the original PDE by applying the differential operator $\mathcal{N}$ to the symbolic expression $u(\hat{\mathbf{x}})$, as defined in Eq. 4. This term ensures that the candidate respects the governing physical laws. The second term, $\mathcal{L}_{\text{MSE2}}$, measures the mean squared error between the predicted and observed values, as shown in Eq. 5. Together, these two components guide the search toward solutions that are both physically valid and numerically precise.

$$\mathcal{L}_{\text{PDE}} = \left\| \mathcal{N} \left( g \left( \hat{u}_x, \hat{u}_y, \hat{u}_z \right) \right) \right\|^2 \tag{4}$$

$$\mathcal{L}_{\text{MSE2}} = \frac{1}{n} \sum_{i=1}^{n} \left( u_i - g \left( \hat{u}_x, \hat{u}_y, \hat{u}_z \right) \right)^2 \tag{5}$$

Table 1: Summary of the model's performance for MSE. (Train,Test)

| Problem | 1-1D | 1-2D | 1-3D | 2-1D | 2-2D | 2-3D |
|---|---|---|---|---|---|---|
| Algorithm | | | Advection | | | |
| NetGP | (**1.31e-15,1.19e-15**) | (**4.59e-15,4.99e-15**) | (4.42e-01,4.24e-01) | (1.43e-11,1.97e-11) | (8.25e-02,8.72e-02) | (4.70e-01,3.92e-01) |
| HD-TLGP | (5.25e-12,5.37e-12) | (4.98e-10,5.06e-10) | (**1.71e-12,1.76e-12**) | (9.92e-03,1.32e-02) | (2.97e-02,4.25e-02) | (5.69e-02,7.64e-02) |
| PINN | (3.51e-01,4.16e-01) | (1.46e+00,1.49e+00) | (3.06e+00,3.59e+00) | (2.10e-01,2.52e-01) | (7.01e-01,9.13e-01) | (1.51e+00,2.03e+00) |
| PSR | (5.83e-11,7.13e-11) | (4.27e-11,3.79e-11) | (1.28e-10,1.42e-10) | (**8.73e-16,7.08e-16**) | (**1.81e-10,2.09e-10**) | (1.51e-03,1.54e-03) |
| PSR+noise | (1.35e-08,2.65e-08) | (1.45e-08,1.34e-08) | (5.47e-10,7.03e-10) | (6.88e-08,1.48e-07) | (2.84e-03,2.79e-03) | (**5.67e-04,5.56e-04**) |
| | | | Diffusion-Reaction | | | |
| NetGP | (5.74e-05,8.06e-05) | (2.14e-03,3.40e-03) | (3.71e-01,2.80e-01) | (1.10e-05,2.77e-05) | (1.57e-03,2.17e-03) | (2.88e-01,3.78e-01) |
| HD-TLGP | (2.20e-02,2.56e-02) | (8.35e-02,1.05e-01) | (5.63e-02,6.14e-02) | (1.27e-02,1.40e-02) | (2.16e-02,2.28e-02) | (4.30e-02,4.08e-02) |
| PINN | (1.59e-02,2.11e-02) | (3.60e-02,7.36e-02) | (6.93e-02,1.63e-01) | (9.43e-03,1.67e-02) | (2.48e-02,5.77e-02) | (4.63e-02,1.27e-01) |
| PSR | (1.92e-06,5.85e-06) | (**1.11e-04,4.14e-04**) | (**3.11e-05,6.51e-05**) | (**4.20e-07,1.29e-06**) | (**1.91e-05,2.53e-05**) | (7.33e-05,3.47e-04) |
| PSR+noise | (**1.56e-08,2.74e-08**) | (1.01e-03,7.60e-04) | (2.21e-04,3.87e-04) | (3.11e-06,2.53e-05) | (1.46e-04,2.86e-04) | (1.17e-03,1.52e-03) |
| | | | Heat | | | |
| NetGP | (2.60e-03,1.66e-02) | (6.20e-03,2.99e+00) | (3.40e-02,1.25e-02) | (6.33e-04,7.11E-03) | (1.09e-03,7.35e-03) | (3.45e-03,**1.20e-03**) |
| HD-TLGP | (1.02e-01,2.65e-02) | (2.09e-02,1.63e-02) | (2.83e+00,3.17e+00) | (5.59e-02,1.48e-02) | (9.00e-03,8.18e-03) | (3.95e-01,3.59e-01) |
| PINN | (1.73e-02,4.81e-02) | (1.98e-02,1.55e-02) | (2.91e-02,**5.39e-03**) | (8.37e-03,2.47e-02) | (4.89e-03,**5.59e-03**) | (2.94e-03,1.21e-03) |
| PSR | (2.57e-03,1.51e-02) | (**4.86e-04,3.23e-03**) | (**9.52e-04**,1.43e-01) | (9.45e-04,8.12e-03) | (2.62e-04,8.03e-01) | (**5.84e-04**,4.26e-03) |
| PSR+noise | (**3.17e-04,3.25e-03**) | (1.53e-03,3.48e-03) | (4.52e-03,5.63e+03) | (**3.40e-04,6.28e-03**) | (**2.01e-04**,1.58e+04) | (5.92e-04,3.88e-03) |
| | | | Poisson | | | |
| NetGP | (3.70e-13,4.92e-13) | (2.34e-02,4.12e-02) | (1.22e-01,1.50e-01) | (6.23e-10,6.86e-10) | (4.55e-04,4.72e-04) | (7.16e-02,1.37e-01) |
| HD-TLGP | (**4.50e-15,4.96e-15**) | (**6.26e-11,6.65e-11**) | (**2.83e-11,2.06e-11**) | (**2.56e-12,2.59e-12**) | (2.07e-11,2.17e-11) | (**2.82e-10,3.14e-10**) |
| PINN | (1.85e-02,2.10e-02) | (6.15e-02,6.32e-02) | (5.10e-02,8.52e-02) | (2.11e-02,2.72e-02) | (9.99e-03,9.15e-03) | (8.81e-03,1.09e-02) |
| PSR | (3.51e-11,3.71e-11) | (6.37e-02,5.56e-02) | (1.13e-02,2.76e-02) | (4.58e-12,7.08e-12) | (**2.62e-12,1.96e-12**) | (4.58e-08,5.98e-08) |
| PSR+noise | (1.90e-08,1.96e-08) | (9.42e-05,1.30e-04) | (2.02e-03,4.72e-03) | (1.65e-08,1.92e-08) | (2.18e-09,2.28e-09) | (5.72e-08,5.68e-08) |
| | | | Wave | | | |
| NetGP | (1.48e-10,1.68e-10) | (3.68e-01,4.33e-01) | (1.25e+00,1.27e+00) | (**1.35e-14**,1.17e-14) | (4.36e-01,4.51e-01) | (4.14e+00,3.84e+00) |
| HD-TLGP | (1.39e-01,1.27e-01) | (2.92e-01,2.50e-01) | (3.81e+00,4.76e+00) | (1.38e-14,**8.42e-15**) | (1.26e+00,1.17e+00) | (2.98e+00,3.06e+00) |
| PINN | (5.09e-01,4.98e-01) | (1.23e+00,1.42e+00) | (2.68e+00,2.82e+00) | (5.50e-01,5.23e-01) | (2.03e+00,2.02e+00) | (4.55e+00,4.47e+00) |
| PSR | (**3.95e-11,3.58e-11**) | (**3.05e-02,3.37e-02**) | (8.36e-01,7.69e-01) | (1.72e-11,1.53e-11) | (2.95e-02,2.74e-01) | (**4.38e-02,4.05e-02**) |
| PSR+noise | (2.56e-09,2.72e-09) | (1.58e-01,1.50e-01) | (**5.18e-01,5.12e-01**) | (4.69e-09,4.17e-09) | (**7.30e-02,8.88e-02**) | (2.03e-01,2.58e-01) |

## 4 EXPERIMENTS

The experiments mainly include four parts. First, numerical data fitting is performed by computing the mean squared error (MSE) between predicted and reference values. Second, PDE fidelity is assessed by substituting the discovered expressions into the governing equations and calculating the PDE residual error. Third, to evaluate interpretability, symbolic solutions obtained by PSR are compared with ground-truth expressions, demonstrating how the underlying physical fields interact across spatial and temporal dimensions. Finally, some ablation studies are conducted to analyze the contribution of each component within the PSR framework.

### 4.1 EXPERIMENT SETTINGS

**PDEs.** To facilitate reproducible evaluation and foster future research, we introduce SymPDEBench, a lightweight benchmark for symbolic regression based PDE solver. It comprises five representative types of PDEs: Advection, Poisson, Heat, Wave, and Diffusion-Reaction. Each PDE includes multiple instances generated under varying coefficients and boundary/initial conditions. Each instance has a known analytical solution. These PDEs span diverse physical phenomena and mathematical characteristics, offering a comprehensive benchmark for assessing performance. A summary of the PDEs is provided in Table 4, with detailed formulations and parameter settings available in the Appendix A.1.

**Dataset Construction.** We sample two separate datasets from the solution domain for training and testing for each PDE instance. We add Gaussian noise with a standard deviation of 0.001 to the training data to simulate real-world conditions where measurements may be imperfect.

**Baselines and Configurations.** We compare the proposed PSR framework against three baselines: NetGP Cao et al. (2025), a state-of-the-art symbolic regression based method originally designed for low-dimensional PDEs; HD-TLGP, a symbolic regression based method specifically designed to tackle high-dimensional PDEs via transfer learning Cao et al. (2024a); and PINN Lu et al. (2021b), a widely used deep learning-based approach for solving PDEs. A comprehensive description of each method, along with detailed parameter settings, is provided in the Appendix A.2.

Table 2: PDE residual error: the MSE of the expression fitting error to the PDE. (Train, Test)

| Problem | 1-1D | 1-2D | 1-3D | 2-1D | 2-2D | 2-3D |
|---|---|---|---|---|---|---|
| Algorithm | | | Advection | | | |
| NetGP | **(0.00e+00,0.00e+00)** | **(0.00e+00,0.00e+00)** | (4.00e+04,4.00e+04) | (1.21e-14,3.01e-13) | (1.26e-05,1.84e-03) | (4.00e+04,4.00e+04) |
| HD-TLGP | (1.82e-14,7.28e-14) | (1.86e-12,7.46e-12) | **(7.11e-15,2.84e-14)** | (1.11e-06,4.43e-06) | (2.83e-06,1.13e-05) | (4.67e-06,1.87e-05) |
| PSR | **(0.00e+00,0.00e+00)** | (9.99e-13,5.00e-13) | (3.13e-13,1.23e-13) | **(0.00e+00,0.00e+00)** | **(4.79e-11,1.54e-12)** | **(7.89e-07,1.13e-07)** |
| PSR+noise | (5.45e-09,6.60e-10) | (1.10e-11,7.90e-11) | (2.38e-12,1.43e-11) | (1.19e-11,9.84e-12) | (2.04e-04,2.35e-05) | (7.75e-05,7.17e-07) |
| | | | Diffusion-Reaction | | | |
| NetGP | **(1.52e-04,2.09e-03)** | **(6.42e-05,**4.62e-02) | (2.50e+05,2.51e+05) | (1.59e-01,1.81e-03) | (2.94e-04,1.74e-02) | (8.97e+04,8.98e+04) |
| HD-TLGP | (1.37e-02,4.71e-03) | (5.63e-02,**2.41e-02**) | (2.55e-03,**1.17e-02**) | **(2.17e-05,**3.74e-03) | (2.14e-02,**2.99e-03**) | (2.57e-02,**5.13e-03**) |
| PSR | (1.58e-04,2.67e-03) | (4.64e-04,2.47e-02) | (1.58e-03,2.49e-01) | (1.48e-01,**1.11e-03**) | **(1.02e-04,**1.26e-02) | **(7.90e-03,**1.24e-01) |
| PSR+noise | (1.62e-04,2.52e-03) | (2.06e-04,4.21e-02) | **(9.93e-04,**2.63e-01) | (1.55e-01,1.32e-03) | (3.37e-04,1.44e-02) | (8.42e-02,9.37e-02) |
| | | | Heat | | | |
| NetGP | (1.44e-03,2.93e-04) | (2.76e-04,2.91e-03) | (4.00e+02,4.00e+02) | (3.41e-04,3.19e-03) | (1.68e-06,1.36e-04) | (1.00e+04,1.00e+04) |
| HD-TLGP | (**1.91e-07,7.72e-08**) | (**7.16e-07,3.33e-09**) | (4.00e+02,3.99e+02) | (3.38e-06,1.37e-06) | **(0.00e+00,0.00e+00)** | (1.00e+04,1.00e+04) |
| PSR | (5.46e-05,1.65e-03) | (4.78e-05,2.68e-03) | (4.80e-06,**3.62e-05**) | (3.94e-04,2.41e-03) | (1.67e-06,3.87e-04) | (6.49e-07,**2.90e-05**) |
| PSR+noise | (1.69e-05,4.21e-04) | (6.79e-05,2.28e-03) | (**4.75e-06,**9.50e-05) | (1.61e-04,6.64e-04) | (2.90e-06,3.26e-05) | (**1.66e-08,**8.24e-05) |
| | | | Poisson | | | |
| NetGP | (**5.11e+01,**4.81e+01) | (**7.07e+01,**9.31e+01) | (4.20e+04,4.30e+04) | (**9.98e-01,**9.99e-01) | (**9.98e-01,**9.99e-01) | (4.04e+04,4.04e+04) |
| HD-TLGP | (5.16e+01,**4.58e+01**) | (1.04e+02,**8.47e+01**) | (1.10e+02,1.07e+02) | (9.99e-01,**9.95e-01**) | (9.99e-01,**9.95e-01**) | (9.99e-01,**9.95e-01**) |
| PSR | (**5.11e+01,**4.81e+01) | (**7.07e+01,**9.31e+01) | (**7.40e+01,**9.91e+01) | (**9.98e-01,**9.99e-01) | (**9.98e-01,**9.99e-01) | (**9.98e-01,**9.99e-01) |
| PSR+noise | (**5.11e+01,**4.81e+01) | (**7.07e+01,**9.31e+01) | (**7.40e+01,**9.91e+01) | (**9.98e-01,**9.99e-01) | (**9.98e-01,**9.99e-01) | (**9.98e-01,**9.99e-01) |
| | | | Wave | | | |
| NetGP | (5.86e-11,3.49e-11) | (1.34e-03,**8.45e-06**) | (1.00e+04,1.00e+04) | **(0.00e+00,0.00e+00)** | (7.28e-15,9.09e-16) | (2.89e+06,2.89e+06) |
| HD-TLGP | **(0.00e+00,0.00e+00)** | (**1.49e-04,**3.31e-05) | (1.00e+04,9.99e+03) | **(0.00e+00,0.00e+00)** | **(0.00e+00,0.00e+00)** | (**2.54e-02,8.43e-01**) |
| PSR | (9.17e-12,5.39e-12) | (6.14e-02,8.48e-02) | (1.00e+04,1.00e+04) | (3.64e-14,2.27e-14) | (9.99e+03,1.00e+04) | (9.99e+03,1.00e+04) |
| PSR+noise | (3.71e-09,1.38e-10) | (5.00e-02,1.06e-01) | (**2.67e-02,8.22e-04**) | (1.22e-10,4.73e-11) | (6.40e+05,6.40e+05) | (6.40e+05,6.40e+05) |

### 4.2 NUMERICAL DATA FITTING COMPARISON

This experiment evaluates the performance of PSR in fitting numerical datasets, both clean and noisy. Table 1 reports the Mean Squared Error (MSE) on training and test sets across various benchmark problems. While NetGP performs adequately in simple 1D cases, it struggles in higher dimensions due to the curse of dimensionality. PINN yields moderate results. HD-TLGP achieves 13 best results, but its overall performance is constrained by its dependence on accurate one-dimensional solutions.

PSR outperforms other baselines, achieving the lowest MSE in 39 out of 60 cases (highlighted in bold), and matching the best order of magnitude in 3 additional cases (highlighted in gray).

Importantly, when Gaussian noise is added to the training data, PSR maintains stable performance on the clean training and test sets. This robustness suggests that PSR fits the data accurately and generalizes well, even under noisy conditions.

Table 3: Comparison of ground-truth and PSR-derived solutions

| Problem | 1D | 2D | 3D |
|---------|-----|-----|-----|
| | | Advection1- | |
| True | $x_1 - t$ | $x_1 + x_2 - 2t$ | $x_1 + x_2 + x_3 - 3t$ |
| PSR | $-1.0 \cdot f_1$ | $(f_2 + (f_1 - (f_1 \cdot 1.0)))$ | $2.0 \cdot f_1 - 1.0 \cdot f_2 + f_3$ |
| | | Advection2- | |
| True | $\sin(x_1 - t)$ | $\sin(x_1 - t) + \sin(x_2 - t)$ | $\sin(x_1 - t) + \sin(x_2 - t) + \sin(x_3 - t)$ |
| PSR | $f_1$ | $-f_1 - f_2 + 2.0$ | $f_1 + f_2 + (f_2 + f_3) \cdot sin(f_2) + 0.84$ |

## 4.3 PARTIAL DIFFERENTIAL EQUATION FITTING COMPARISON

This experiment evaluates the proposed PSR framework's ability to discover solutions that are consistent with the underlying physical laws, particularly as the dimensionality of the problem increases. Table 2 presents a comparison of the final PDE residual errors across three methods: NetGP, HD-TLGP, and PSR. NetGP achieves the lowest residuals in 13 cases, while HD-TLGP leads in 29 cases. PSR obtains the best results in 25 cases and matches the best order of magnitude in 7 additional ones. These results demonstrate that PSR achieves strong physical consistency and exhibits robust performance, even when trained on noisy data.

## 4.4 REVEALING THE DEEPER PHYSICAL INSIGHT

A key claim of our work is that PSR offers deeper physical insight by revealing how the physical field governed by the PDE interacts across its various dimensions. To demonstrate this, we compare the high-level compositional structure discovered by our global SR stage with the ground-truth in Table 3 (full results provided in the Appendix C), where $f_1, f_2, f_3$ represent the functional components discovered from the projections. The results show that PSR can discover the correct compositional form of lower-dimensional functions (such as the Advection).

## 4.5 ABLATION STUDIES

To validate the key components of our proposed framework, we conduct a series of ablation studies. First, we demonstrate the effectiveness of PSR as an incremental builder that can enhance existing algorithms. As shown in Table 1 and Table 2, PSR is incorporated into NetGP and consistently improves accuracy and generalization across all high-dimensional PDE tasks, confirming the benefit of this incremental builder. Next, we conduct an ablation study on the Heat 2-3D problem to isolate the contribution of our decomposition strategy, which is specifically designed for PDEs. The results are summarized in Table 10 and Appendix D. Finally, we conduct an ablation experiment on the Heat 2-1D problem to assess the effectiveness of the introduced MSEDI loss. Figure 4 shows that "MSE + MSEDI" performs comparably to "MSE + PDE residual error", indicating that MSEDI effectively captures local dynamics of the solution without computing the full PDE residual error.

## 5 CONCLUSION

In this paper, we introduced Projective Symbolic Regression (PSR) to discover interpretable solutions for high-dimensional PDEs by decomposing the search into low-dimensional projections and composing functional components hierarchically. Experiments on high-dimensional PDEs demonstrate that PSR successfully discovers accurate and interpretable symbolic solutions that reveal the underlying compositional structure of the physics.

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

## A    DETAILED EXPERIMENTAL SETUP

### A.1    PROBLEM DESCRIPTION

Table 4: Summary of PDE

| Name | Application | General form |
|------|-------------|--------------|
| Advection | Transport physical quantity | $\frac{\partial \phi}{\partial t} + \mathbf{v} \cdot \nabla \phi = 0$ |
| Poisson | Potential field | $\Delta \phi = f$ |
| Heat | Heat (or temperature) | $\Delta \phi = -\frac{q}{k}$ |
| Wave | Propagation of waves | $\frac{\partial^2 u}{\partial t^2} = c^2 \nabla^2 u$ |
| Diffusion-reaction | Chemical Concentration | $\frac{\partial u}{\partial t} = D\nabla^2 u + R(u)$ |

A detailed description of the PDEs addressed in the experiment is presented.

**Advection equation**, also known as the transport equation, models the evolution of a physical quantity as it is carried by a moving fluid. Eq. 6 shows the general form of advection equation, where $\phi$ is the physical quantity being transported (e.g., temperature, concentration), $t$ is time, $\mathbf{v}$ is the velocity vector of the fluid, $\nabla \phi$ is the gradient of $\phi$.

$$\frac{\partial \phi}{\partial t} + \mathbf{v} \cdot \nabla \phi = 0 \tag{6}$$

**Diffusion-reaction equation** models the behavior of chemical substances undergoing diffusion and reaction processes. Eq. 7 shows the general form of diffusion-reaction equation, where $u = u(x_1, x_2, \ldots, x_n, t)$ is the concentration of the substance, $t$ is time, $D$ is the diffusion coefficient, $\nabla^2$ is the Laplacian operator, $R(u)$ is the reaction term, which can be a function of $u$.

$$\frac{\partial u}{\partial t} = D\nabla^2 u + R(u) \tag{7}$$

**Heat equation** models how heat (or temperature) diffuses through a given region over time. Eq. 8 shows the general form of the heat equation, where $u$ is the temperature, $\alpha$ is the thermal diffusivity of the material. The diffusivity is defined as $\alpha = \frac{k}{\rho c}$, where $k$ denotes the thermal conductivity, $\rho$ is the density, and $c$ is the specific heat capacity.

$$\frac{\partial u}{\partial t} = \alpha \Delta u \tag{8}$$

**Poisson equation** models the potential field generated by the source term. Eq. 9 shows the general form of the poisson equation, where $\Delta$ is the Laplace operator, often denoted as $\nabla^2$. $\phi$ is the unknown function. $f$ is a known function, typically representing a source term or distribution.

$$\Delta \phi = f \tag{9}$$

**Wave equation** models the propagation of waves, such as sound, light, and water waves. Eq. 10 shows the general form of wave equation, where $u = u(x_1, x_2, \ldots, x_n, t)$ is the wave function, $t$ is time, $\nabla^2$ is the Laplacian operator, $c$ is the speed of wave propagation.

$$\frac{\partial^2 u}{\partial t^2} = c^2 \nabla^2 u \tag{10}$$

Table 5 assigns a simple name to each PDE and provides detailed specifications. The notation $\partial \cdot$ denotes the boundary of the domain $\cdot$.

Table 5: Summary of PDE with their initial condition, boundary condition, and analytical solution.

| Name | PDE | Initial Condition | Boundary Condition | Set of Operators |
|---|---|---|---|---|
| Advection 1-1D | $\frac{\partial u}{\partial t} + \frac{\partial u}{\partial x_1} = 0, \mathbf{x} \in [0,1], t \in (0,2]$ | $u(\mathbf{x},0) = x_1$ | $u(0,t) = -t, u(1,t) = 1-t$ | $\times, +, -$ |
| Advection 1-2D | $\frac{\partial u}{\partial t} + \frac{\partial u}{\partial x_1} + \frac{\partial u}{\partial x_2} = 0, \mathbf{x} \in [0,1]^2, t \in (0,2]$ | $u(\mathbf{x},0) = x_1 + x_2$ | - | $\times, +, -$ |
| Advection 1-3D | $\frac{\partial u}{\partial t} + \frac{\partial u}{\partial x_1} + \frac{\partial u}{\partial x_2} + \frac{\partial u}{\partial x_3} = 0, \mathbf{x} \in [0,1]^3, t \in (0,2]$ | $u(\mathbf{x},0) = x_1 + x_2 + x_3$ | - | $\times, +, -$ |
| Advection 2-1D | $\frac{\partial u}{\partial t} + \frac{\partial u}{\partial x_1} = 0, \mathbf{x} \in [0,1], t \in (0,2]$ | $u(\mathbf{x},0) = \sin(x_1)$ | $u(0,t) = \sin(-t), u(1,t) = \sin(1-t)$ | $\times, +, -, \sin$ |
| Advection 2-2D | $\frac{\partial u}{\partial t} + \frac{\partial u}{\partial x_1} + \frac{\partial u}{\partial x_2} = 0, \mathbf{x} \in [0,1]^2, t \in (0,2]$ | $u(\mathbf{x},0) = \sin(x_1) + \sin(x_2)$ | - | $\times, +, -, \sin$ |
| Advection 2-3D | $\frac{\partial u}{\partial t} + \frac{\partial u}{\partial x_1} + \frac{\partial u}{\partial x_2} + \frac{\partial u}{\partial x_3} = 0, \mathbf{x} \in [0,1]^3, t \in (0,2]$ | $u(\mathbf{x},0) = \sin(x_1) + \sin(x_2) + \sin(x_3)$ | - | $\times, +, -, \sin$ |
| Diffusion-reaction 1-1D | $\frac{\partial u}{\partial t} - 3(\frac{\partial^2 u}{\partial x_1^2}) - u = 0, \mathbf{x} \in [0,1], t \in (0,2]$ | $u(\mathbf{x},0) = \sin(x_1)$ | - | $\times, +, -, \sin, \exp$ |
| Diffusion-reaction 1-2D | $\frac{\partial u}{\partial t} - 3(\frac{\partial^2 u}{\partial x_1^2} + \frac{\partial^2 u}{\partial x_2^2}) - u = 0, \mathbf{x} \in [0,1]^2, t \in (0,2]$ | $u(\mathbf{x},0) = \sin(x_1) + \sin(x_2)$ | - | $\times, +, -, \sin, \exp$ |
| Diffusion-reaction 1-3D | $\frac{\partial u}{\partial t} - 3(\frac{\partial^2 u}{\partial x_1^2} + \frac{\partial^2 u}{\partial x_2^2} + \frac{\partial^2 u}{\partial x_3^2}) - u = 0, \mathbf{x} \in [0,1]^3, t \in (0,2]$ | $u(\mathbf{x},0) = \sin(x_1) + \sin(x_2) + \sin(x_3)$ | - | $\times, +, -, \sin, \exp$ |
| Diffusion-reaction 2-1D | $\frac{\partial u}{\partial t} - 2(\frac{\partial^2 u}{\partial x_1^2}) + u = 0, \mathbf{x} \in [0,1], t \in (0,2]$ | $u(\mathbf{x},0) = \sin(x_1)$ | - | $\times, +, -, \sin, \exp$ |
| Diffusion-reaction 2-2D | $\frac{\partial u}{\partial t} - 2(\frac{\partial^2 u}{\partial x_1^2} + \frac{\partial^2 u}{\partial x_2^2}) + u = 0, \mathbf{x} \in [0,1]^2, t \in (0,2]$ | $u(\mathbf{x},0) = \sin(x_1) + \sin(x_2)$ | - | $\times, +, -, \sin, \exp$ |
| Diffusion-reaction 2-3D | $\frac{\partial u}{\partial t} - 2(\frac{\partial^2 u}{\partial x_1^2} + \frac{\partial^2 u}{\partial x_2^2} + \frac{\partial^2 u}{\partial x_3^2}) + u = 0, \mathbf{x} \in [0,1]^3, t \in (0,2]$ | $u(\mathbf{x},0) = \sin(x_1) + \sin(x_2) + \sin(x_3)$ | - | $\times, +, -, \sin, \exp$ |
| Heat 1-1D | $\frac{\partial u}{\partial t} - 0.4\left(\frac{\partial^2 u}{\partial x_1^2}\right) = 0, \mathbf{x} \in [0,1], t \in (0,2]$ | $u(\mathbf{x},0) = \delta(\mathbf{x})$ | - | $\times, +, -, \exp, \sqrt{\cdot}, \div$ |
| Heat 1-2D | $\frac{\partial u}{\partial t} - 0.4\left(\frac{\partial^2 u}{\partial x_1^2} + \frac{\partial^2 u}{\partial x_2^2}\right) = 0, \mathbf{x} \in [0,1]^2, t \in (0,2]$ | $u(\mathbf{x},0) = \delta(\mathbf{x})$ | - | $\times, +, -, \exp, \sqrt{\cdot}, \div$ |
| Heat 1-3D | $\frac{\partial u}{\partial t} - 0.4\left(\frac{\partial^2 u}{\partial x_1^2} + \frac{\partial^2 u}{\partial x_2^2} + \frac{\partial^2 u}{\partial x_3^2}\right) = 0, \mathbf{x} \in [0,1]^3, t \in (0,2]$ | $u(\mathbf{x},0) = \delta(\mathbf{x})$ | - | $\times, +, -, \exp, \sqrt{\cdot}, \div$ |
| Heat 2-1D | $\frac{\partial u}{\partial t} - \left(\frac{\partial^2 u}{\partial x_1^2}\right) = 0, \mathbf{x} \in [0,1], t \in (0,2]$ | $u(\mathbf{x},0) = \delta(\mathbf{x})$ | - | $\times, +, -, \exp, \sqrt{\cdot}, \div$ |
| Heat 2-2D | $\frac{\partial u}{\partial t} - \left(\frac{\partial^2 u}{\partial x_1^2} + \frac{\partial^2 u}{\partial x_2^2}\right) = 0, \mathbf{x} \in [0,1]^2, t \in (0,2]$ | $u(\mathbf{x},0) = \delta(\mathbf{x})$ | - | $\times, +, -, \exp, \sqrt{\cdot}, \div$ |
| Heat 2-3D | $\frac{\partial u}{\partial t} - \left(\frac{\partial^2 u}{\partial x_1^2} + \frac{\partial^2 u}{\partial x_2^2} + \frac{\partial^2 u}{\partial x_3^2}\right) = 0, \mathbf{x} \in [0,1]^3, t \in (0,2]$ | $u(\mathbf{x},0) = \delta(\mathbf{x})$ | - | $\times, +, -, \exp, \sqrt{\cdot}, \div$ |
| Poisson 1-1D | $\frac{\partial^2 u}{\partial x_1^2} + \pi^2 \sin(\pi x_1) = 0, \mathbf{x} \in [0,1]$ | - | $u(\mathbf{x}) = 0, \mathbf{x} \in \partial[0,1]$ | $\times, +, -, \sin$ |
| Poisson 1-2D | $\frac{\partial^2 u}{\partial x_1^2} + \frac{\partial^2 u}{\partial x_2^2} + 2\pi^2 \sin(\pi x_1)\sin(\pi x_2) = 0, \mathbf{x} \in [0,1]^2$ | - | $u(\mathbf{x}) = 0, \mathbf{x} \in \partial[0,1]^2$ | $\times, +, -, \sin$ |
| Poisson 1-3D | $\frac{\partial^2 u}{\partial x_1^2} + \frac{\partial^2 u}{\partial x_2^2} + \frac{\partial^2 u}{\partial x_3^2} + 3\pi^2 \sin(\pi x_1)\sin(\pi x_2)\sin(\pi x_3) = 0, \mathbf{x} \in [0,1]^3$ | - | $u(\mathbf{x}) = 0, \mathbf{x} \in \partial[0,1]^3$ | $\times, +, -, \sin$ |
| Poisson 2-1D | $\frac{\partial^2 u}{\partial x_1^2} + 1 = 0, \mathbf{x} \in \{x_1 \mid x_1^2 \leq 1\}$ | - | $u(\mathbf{x}) = 0, \mathbf{x} \in \{x_1 \mid x_1^2 = 1\}$ | $\times, +, -, \sin$ |
| Poisson 2-2D | $\frac{\partial^2 u}{\partial x_1^2} + \frac{\partial^2 u}{\partial x_2^2} + 1 = 0, \mathbf{x} \in \{(x_1, x_2) \mid x_1^2 + x_2^2 \leq 1\}$ | - | $u(\mathbf{x}) = 0, \mathbf{x} \in \{(x_1, x_2) \mid x_1^2 + x_2^2 = 1\}$ | $\times, +, -, \sin$ |
| Poisson 2-3D | $\frac{\partial^2 u}{\partial x_1^2} + \frac{\partial^2 u}{\partial x_2^2} + \frac{\partial^2 u}{\partial x_3^2} + 1 = 0, \mathbf{x} \in \{(x_1, x_2, x_3) \mid x_1^2 + x_2^2 + x_3^2 \leq 1\}$ | - | $u(\mathbf{x}) = 0, \mathbf{x} \in \{(x_1, x_2, x_3) \mid x_1^2 + x_2^2 + x_3^2 = 1\}$ | $\times, +, -, \sin$ |
| Wave 1-1D | $\frac{\partial^2 u}{\partial t^2} - 1^2(\frac{\partial^2 u}{\partial x_1^2}) = 0, \mathbf{x} \in [0,1], t \in (0,2]$ | $u(\mathbf{x},0) = \sin(3x_1)$ | - | $\times, +, \sin$ |
| Wave 1-2D | $\frac{\partial^2 u}{\partial t^2} - 1^2(\frac{\partial^2 u}{\partial x_1^2} + \frac{\partial^2 u}{\partial x_2^2}) = 0, \mathbf{x} \in [0,1]^2, t \in (0,2]$ | $u(\mathbf{x},0) = \sin(3x_1) + \sin(3x_2)$ | - | $\times, +, \sin$ |
| Wave 1-3D | $\frac{\partial^2 u}{\partial t^2} - 1^2(\frac{\partial^2 u}{\partial x_1^2} + \frac{\partial^2 u}{\partial x_2^2} + \frac{\partial^2 u}{\partial x_3^2}) = 0, \mathbf{x} \in [0,1]^3, t \in (0,2]$ | $u(\mathbf{x},0) = \sin(3x_1) + \sin(3x_2) + \sin(3x_3)$ | - | $\times, +, \sin$ |
| Wave 2-1D | $\frac{\partial^2 u}{\partial t^2} - 3^2(\frac{\partial^2 u}{\partial x_1^2}) = 0, \mathbf{x} \in [0,1], t \in (0,2]$ | $u(\mathbf{x},0) = \sin(x_1)$ | - | $\times, +, \sin$ |
| Wave 2-2D | $\frac{\partial^2 u}{\partial t^2} - 3^2(\frac{\partial^2 u}{\partial x_1^2} + \frac{\partial^2 u}{\partial x_2^2}) = 0, \mathbf{x} \in [0,1]^2, t \in (0,2]$ | $u(\mathbf{x},0) = \sin(x_1) + \sin(x_2)$ | - | $\times, +, \sin$ |
| Wave 2-3D | $\frac{\partial^2 u}{\partial t^2} - 3^2(\frac{\partial^2 u}{\partial x_1^2} + \frac{\partial^2 u}{\partial x_2^2} + \frac{\partial^2 u}{\partial x_3^2}) = 0, \mathbf{x} \in [0,1]^3, t \in (0,2]$ | $u(\mathbf{x},0) = \sin(x_1) + \sin(x_2) + \sin(x_3)$ | - | $\times, +, \sin$ |

## A.2 PARAMETERS SETTING

Table 6 presents the key parameter settings for the PSR. HD-TLGP and NetGP share the same parameters with PSR. The termination condition is defined by a reward value exceeding 0.999, where the reward function is formulated as Eq. 11 or Eq. 12. These reward functions yield values in the range $(0, 1]$ with values closer to 1 indicates better performance.

A fair evaluation requires that each baseline is tested under its optimal conditions. The performance of the HD-TLGP method is directly dependent on the quality of the one-dimensional solutions it uses for transfer learning, which are generated by NetGP. Compared to its performance reported in the original paper, NetGP achieves significantly better results when trained on randomly sampled data rather than boundary-sampled data. Therefore, to ensure that HD-TLGP operates under its most favorable conditions and its performance is not unfairly compromised by a suboptimal data sampling strategy, we use randomly sampled datasets for both training and evaluating HD-TLGP.

$$R = \frac{1}{0.001 \cdot \mathcal{L}_{MSE1} + \mathcal{L}_{MSEDI} + 1} \tag{11}$$

$$R = \frac{1}{0.001 \cdot \mathcal{L}_{MSE2} + \mathcal{L}_{PDE} + 1} \tag{12}$$

Table 6: Main Parameters Setting of PSR.

| | |
|---|---|
| Default operators | $\times, +, -, \sin, \exp, \sqrt{\cdot}, \div$ |
| Set of terminals | $x_1, x_2, x_3, t, c$ |
| Population Size | 100 |
| Maximum generations for local SR | 10 |
| Maximum generations for global SR | 5 |
| Keep for the next generation (nkeep) | 10 |
| Termination condition | $> 0.999$ |
| Crossover probability | 0.6 |
| Mutation probability | 0.6 |
| System configuration | Intel(R) Core(TM) i5-1135G7 CPU |

Table 7 presents the key parameter settings for the PINN. The Physics-Informed Neural Network (PINN) is trained using a two-stage optimization strategy. The first stage employs the Adam optimizer with a learning rate of $1e - 04$ and runs for 10,000 iterations. This phase ensures initial convergence and balances the data-driven and physics-based loss components. In the second stage, the model is fine-tuned using the L-BFGS optimizer, which internally determines the number of iterations based

Table 7: Main Parameters Setting of PINN.

| | |
|---|---|
| Optimizer (Phase 1) | adam |
| Learning Rate | 1e-4 |
| Iterations 1 | 10000 |
| Optimizer (Phase 2) | L-BFGS (optimizer for fine-tuning) |
| Iterations 2 | Determined internally by L-BFGS convergence |
| System configuration | NVIDIA GeForce RTX 3090 GPU |
| Framework | DeepXDE |

Table 8: Summary of the Model's Performance for MSE Trained on Noisy Dataset.

| Problem | Train (Noisy) | | | Clean Training Datasets | | | Test | | |
|---|---|---|---|---|---|---|---|---|---|
| | NetGP | PINN | PSR | NetGP | PINN | PSR | NetGP | PINN | PSR |
| Advection 1-1D | 1.08e-06 | 3.21e-01 | **1.07e-06** | **5.46e-09** | 3.53e-01 | 1.35e-08 | **9.53e-09** | 4.13e-01 | 2.65e-08 |
| Advection 1-2D | 9.55e-07 | 1.32e+00 | **9.53e-07** | **1.07e-08** | 1.46e+00 | 1.45e-08 | **9.24e-09** | 1.54e+00 | 1.34e-08 |
| Advection 1-3D | 4.40e-01 | 3.13e+00 | **9.97e-07** | 4.40e-01 | 3.06e+00 | **5.47e-10** | 4.14e-01 | 3.50e+00 | **7.03e-10** |
| Advection 2-1D | **9.47e-07** | 2.30e-01 | 9.88e-07 | **1.42e-09** | 2.12e-01 | 6.88e-08 | **2.05e-09** | 2.48e-01 | 1.48e-07 |
| Advection 2-2D | 1.52e-02 | 7.05e-01 | **2.31e-03** | 1.52e-02 | 7.01e-01 | **2.84e-03** | 1.36e-02 | 9.12e-01 | **2.79e-03** |
| Advection 2-3D | 8.27e-01 | 1.48e+00 | **5.33e-04** | 8.27e-01 | 1.52e+00 | **5.67e-04** | 1.01e+00 | 2.10e+00 | **5.56e-04** |
| Diffusion* 1-1D | 5.23e-05 | 1.37e-02 | **1.01e-06** | 6.14e-05 | 1.58e-02 | **1.56e-08** | 7.94e-05 | 2.12e-02 | **2.74e-08** |
| Diffusion* 1-2D | 2.36e-03 | 4.07e-02 | **6.81e-04** | 3.09e-03 | 3.59e-02 | **1.01e-03** | 2.76e-03 | 7.25e-02 | **7.60e-04** |
| Diffusion* 1-3D | 4.02e-01 | 8.62e-02 | **2.63e-04** | 3.72e-01 | 6.95e-02 | **2.21e-04** | 2.85e-01 | 1.58e-01 | **3.87e-04** |
| Diffusion* 2-1D | **1.18e-06** | 8.57e-03 | 4.22e-06 | **2.10e-09** | 9.44e-03 | 3.11e-06 | **1.95e-09** | 1.67e-02 | 2.53e-05 |
| Diffusion* 2-2D | 7.24e-04 | 2.54e-02 | **1.50e-04** | 6.63e-04 | 2.48e-02 | **1.46e-04** | 1.34e-03 | 5.71e-02 | **2.86e-04** |
| Diffusion* 2-3D | 1.53e-01 | 5.56e-02 | **8.36e-04** | 1.56e-01 | 4.63e-02 | **1.17e-03** | 2.55e-01 | 1.26e-01 | **1.52e-03** |
| Heat 1-1D | 1.78e-03 | 1.80e-02 | **2.90e-04** | 1.89e-03 | 1.73e-02 | **3.17e-04** | 2.60e-02 | 4.83e-02 | **3.25e-03** |
| Heat 1-2D | 8.50e-03 | 1.54e-02 | **1.56e-03** | 9.02e-03 | 1.99e-02 | **1.53e-03** | 1.04e-02 | 1.53e-02 | **3.48e-03** |
| Heat 1-3D | 3.48e-02 | 2.19e-02 | **4.70e-03** | 3.86e-02 | 2.91e-02 | **4.52e-03** | 1.56e-02 | **5.27e-03** | 5.63e+03 |
| Heat 2-1D | 6.49e-04 | 7.38e-03 | **3.95e-04** | 7.83e-04 | 8.39e-03 | **3.40e-04** | 8.38e-03 | 2.49e-02 | **6.28e-03** |
| Heat 2-2D | 1.13e-03 | 3.65e-03 | **1.11e-04** | 1.36e-03 | 4.90e-03 | **2.01e-04** | 1.51e+00 | **5.60e-03** | 1.58e+04 |
| Heat 2-3D | 9.98e-03 | 2.36e-03 | **3.47e-04** | 1.06e-02 | 2.94e-03 | **5.92e-04** | 1.14e-02 | **1.22e-03** | 3.88e-03 |
| Poisson 1-1D | **9.95e-07** | 1.88e-02 | **9.95e-07** | 2.10e-08 | 1.94e-02 | **1.90e-08** | 2.17e-08 | 2.18e-02 | **1.96e-08** |
| Poisson 1-2D | 5.23e-02 | 5.57e-02 | **9.23e-05** | 5.15e-02 | 5.41e-02 | **9.42e-05** | 4.28e-02 | 5.47e-02 | **1.30e-04** |
| Poisson 1-3D | 1.99e-01 | 5.52e-02 | **1.89e-03** | 1.84e-01 | 5.38e-02 | **2.02e-03** | 2.61e-01 | 9.30e-02 | **4.72e-03** |
| Poisson 2-1D | **9.46e-07** | 2.21e-02 | 9.51e-07 | 1.86e-08 | 2.11e-02 | **1.65e-08** | 1.72e-08 | 2.73e-02 | 1.92e-08 |
| Poisson 2-2D | 4.71e-04 | 1.05e-02 | **9.29e-07** | 4.76e-04 | 1.04e-02 | **2.18e-09** | 4.32e-04 | 9.65e-03 | **2.28e-09** |
| Poisson 2-3D | 2.36e-01 | 9.16e-03 | **1.07e-06** | 2.31e-01 | 9.02e-03 | **5.72e-08** | 3.42e-01 | 1.18e-02 | **5.68e-08** |
| Wave 1-1D | **1.02e-06** | 4.68e-01 | **1.02e-06** | **1.67e-09** | 5.10e-01 | 2.56e-09 | **1.62e-09** | 5.03e-01 | 2.72e-09 |
| Wave 1-2D | 2.90e-01 | 1.28e+00 | **1.33e-01** | 2.84e-01 | 1.23e+00 | **1.58e-01** | 3.08e-01 | 1.41e+00 | **1.50e-01** |
| Wave 1-3D | 2.66e+00 | 2.70e+00 | **4.93e-01** | 2.74e+00 | 2.68e+00 | **5.18e-01** | 2.90e+00 | 2.80e+00 | **5.12e-01** |
| Wave 2-1D | **1.05e-06** | 5.28e-01 | **1.05e-06** | 4.92e-09 | 5.50e-01 | **4.69e-09** | 4.45e-09 | 5.25e-01 | **4.17e-09** |
| Wave 2-2D | 5.25e-01 | 2.01e+00 | **6.40e-02** | 5.45e-01 | 2.05e+00 | **7.30e-02** | 5.26e-01 | 2.05e+00 | **8.88e-02** |
| Wave 2-3D | 7.54e+00 | 4.64e+00 | **1.91e-01** | 7.05e+00 | 4.57e+00 | **2.03e-01** | 6.69e+00 | 4.52e+00 | **2.58e-01** |

on convergence criteria. All experiments are conducted using the DeepXDE library on an NVIDIA GeForce RTX 3090 GPU.

## B  DETAILED NUMERICAL DATA FITTING COMPARISON

To further investigate the robustness of the models when trained on noisy datasets, we conducted a comprehensive evaluation across multiple PDE benchmarks. Each model was trained on data with noise and then assessed on three aspects: the noisy training data, the original clean training data, and randomly sampled test points. As shown in Table 8, PSR consistently achieves the lowest MSE across all tasks and evaluation settings, demonstrating exceptional resilience to noise and strong generalization.

## C  DETAILED REVEALING THE DEEPER PHYSICAL INSIGHT

A key claim of our work is that PSR offers deeper physical insight by revealing how the physical field governed by the PDE interacts across its various dimensions. To demonstrate this, we compare the high-level compositional structure discovered by our global SR stage with the ground-truth in Table 9, where $f_1, f_2, f_3$ represent the functional components discovered from the projections.

The results shows that for PDEs whose solutions are fundamentally additive or multiplicative combinations of lower-dimensional functions (such as the Advection, Diffusion-Reaction, and Wave

equations), PSR consistently discovers the correct compositional form. For example, in the 1-3D Advection problem, the true solution is an addition of variables. PSR correctly identifies this additive structure, discovering a high-level solution of the form $2.0 \cdot f_1 - 1.0 \cdot f_2 + f_3$.

While the discovered high-level expressions are not always a perfect one-to-one match with the simplest form of the true solution, they correctly capture the essential nature of the interaction between the dimensional components (e.g., whether they add or multiply). This ability to uncover the underlying compositional grammar of a physical system is a unique advantage of our hierarchical approach, providing a level of interpretability that is unattainable with black-box models.

Table 9: Comparison of ground-truth and PSR-derived solutions

| Problem | 1D | 2D | 3D |
|---|---|---|---|
| | | Advection1- | |
| True | $x_1 - t$ | $x_1 + x_2 - 2t$ | $x_1 + x_2 + x_3 - 3t$ |
| PSR | $-1.0 \cdot f_1$ | $(f_2 + (f_1 - (f_1 \cdot 1.0)))$ | $2.0 \cdot f_1 - 1.0 \cdot f_2 + f_3$ |
| | | Advection2- | |
| True | $\sin(x_1 - t)$ | $\sin(x_1 - t) + \sin(x_2 - t)$ | $\sin(x_1 - t) + \sin(x_2 - t) + \sin(x_3 - t)$ |
| PSR | $f_1$ | $-f_1 - f_2 + 2.0$ | $f_1 + f_2 + (f_2 + f_3) \cdot \sin(f_2) + 0.84$ |
| | | Diffusion-Reaction1- | |
| True | $e^{-2t}\sin(x_1)$ | $e^{-2t}\sin(x_1) + e^{-2t}\sin(x_2)$ | $e^{-2t}\sin(x_1) + e^{-2t}\sin(x_2) + e^{-2t}\sin(x_3)$ |
| PSR | $2.0 \cdot f_1 + f_1 \cdot exp(-2 \cdot f_1)$ | $(f_2 - (f_1 + 1.0)(f_1 - f_2(2 \cdot f_2 + 1.0)))exp(f_2)$ | $-f_1 + 2 \cdot f_2 + f_3 + 1.0$ |
| | | Diffusion-Reaction2- | |
| True | $e^{-3t}\sin(x_1)$ | $e^{-3t}\sin(x_1) + e^{-3t}\sin(x_2)$ | $e^{-3t}\sin(x_1) + e^{-3t}\sin(x_2) + e^{-3t}\sin(x_3)$ |
| PSR | $f_1(1.0 \cdot f_1 - 1.0 \cdot \sin(f_1) + 1.84)$ | $f_1\sin(2.71exp(f_1 - f_2))$ | $f_1 \cdot exp(f_1) + f_2 + f_3$ |
| | | Heat1- | |
| True | $\frac{1}{\sqrt{\pi \cdot 1.6 \cdot t}}e^{-\frac{x_1^2}{1.6t}}$ | $\frac{1}{4\pi \times 0.4t}e^{-\frac{x_1^2 + x_2^2}{4 \times 0.4t}}$ | $\frac{1}{(4\pi \times 0.4t)^{3/2}}e^{-\frac{x_1^2 + x_2^2 + x_3^2}{4 \times 0.4t}}$ |
| PSR | $f_1^{-1.75} - 1.0/f_1 + 1.0$ | $f_1 \cdot (1.0 \cdot f_1 \cdot f_2 - f_1 + 1.0)$ | $-f_2 \cdot (f_1 + 2 \cdot f_2 + 2 \cdot f_3 - 1.0)/(f_1 \cdot f_3)$ |
| | | Heat2- | |
| True | $\frac{1}{\sqrt{\pi \cdot 4 \cdot t}}e^{-\frac{x_1^2}{4t}}$ | $\frac{1}{4\pi t}e^{-\frac{x_1^2 + x_2^2}{4t}}$ | $\frac{1}{(4\pi t)^{3/2}}e^{-\frac{x_1^2 + x_2^2 + x_3^2}{4t}}$ |
| PSR | $-1.0 \cdot f_1$ | $f_1 + 1.0 \cdot f_2$ | $2 \cdot f_3/(f_1 + f_2 + exp(f_1))$ |
| | | Poisson1- | |
| True | $\sin(\pi x_1)$ | $\sin(\pi x_1)\sin(\pi x_2)$ | $\sin(\pi x_1)\sin(\pi x_2)\sin(\pi x_3)$ |
| PSR | $-1.0 \cdot f_1 - 0.158$ | $-f_2 + \sin(f_1 - 1.0)$ | $-f_2 + f_3 \cdot (f_3 - 1.0)\sin(f_1) - f_3$ |
| | | Poisson2- | |
| True | $\sin(\pi x_1)$ | $\sin(\pi x_1)\sin(\pi x_2)$ | $\sin(\pi x_1)\sin(\pi x_2)\sin(\pi x_3)$ |
| PSR | $-1.0 \cdot f_1 - 0.158$ | $-f_2 + \sin(f_1 - 1.0)$ | $-f_2 + f_3 \cdot (f_3 - 1.0)\sin(f_1) - f_3$ |
| | | Wave1- | |
| True | $sin(3x_1 + 3t)$ | $sin(3x_1 + 3t) + sin(3x_2 + 3t)$ | $sin(3x_1 + 3t) + sin(3x_2 + 3t) + sin(3x_3 + 3t)$ |
| PSR | $0.841 \cdot f_1$ | $f_2 + \sin(2 \cdot f_1 \cdot f_2 + f_1)$ | $f_1 + \sin(f_1 \cdot f_2 + f_3) + 1.0$ |
| | | Wave2- | |
| True | $sin(x_1 + 3t)$ | $sin(x_1 + 3t) + sin(x_2 + 3t)$ | $sin(x_1 + 3t) + sin(x_2 + 3t) + sin(x_3 + 3t)$ |
| PSR | $2 \cdot f_1 + 3.0$ | $2.0 \cdot f_1^2 \cdot f_2$ | $f_1 \cdot f_2 + 3.0 \cdot f_3$ |

## D    DETAILED ABLATION STUDIES

We first discuss the performance improvement. The proposed PSR framework is designed as an incremental builder that can be integrated into any SR algorithm. As shown in Table 1 and Table 2, PSR is incorporated into NetGP and consistently improves accuracy and generalization across all high-dimensional PDE tasks, confirming the benefit of this incremental builder.

We then conduct an ablation experiment on the Heat 2-3D to prove the effective of the decomposition strategy specified designed for PDE in Table 10. The first row corresponds to the decomposition strategy generated by our method, while the remaining rows represent randomly selected decomposition strategies. Our approach achieves the lowest PDE residual error on both the training and test sets. It also obtains the lowest data fitting error on the training set and matches the best order of magnitude on the test set.

Finally, we conduct an ablation experiment on the Heat 2-1D problem to assess the effectiveness of the introduced MSEDI loss. Figure 4 shows that "MSE + MSEDI" performs comparably to "MSE + PDE residual error", indicating that MSEDI effectively captures local dynamics of the solution without computing the full PDE residual error.

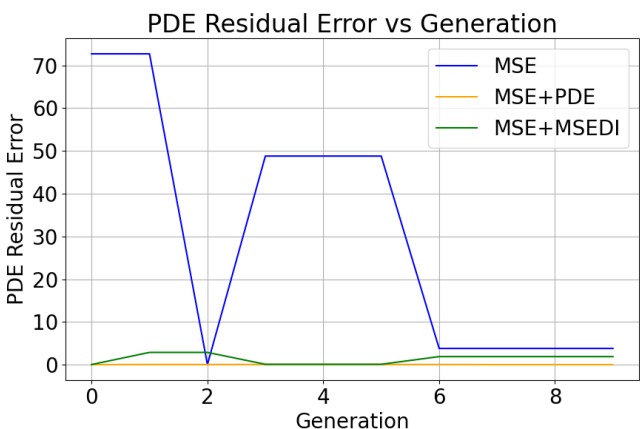

Figure 4: Evaluating the effectiveness of MSEDI on heat 2-1D.

Table 10: Projection Strategy Comparison on Heat 2-3D.

| Projection strategy | Data fitting error (Train, Test) | PDE residual error (Train, Test) |
|---|---|---|
| $(x_1,t), (x_2,t), (x_3,t)$ (Ours) | (**5.84e-04**, 4.26e-03 ) | (**6.49e-07**, **2.90e-05**) |
| $(x_1,t), (x_2), (x_3)$ | (3.27e-03, 5.29e-03) | (4.00e+04, 3.99e+04) |
| $(x_2,t), (x_1), (x_3)$ | (2.88e-03, **1.15e-03**) | (9.99e+03, 9.99e+03) |
| $(x_3,t), (x_1), (x_2)$ | (1.20e-01, 1.07e-01) | (9.99e+03, 9.99e+03) |

## E    ETHICS STATEMENT

All authors of this submission confirm that they have read and agree to abide by the ICLR Code of Ethics.

## F    REPRODUCIBILITY STATEMENT

All data and source code are provided in the supplementary materials to support reproducibility of our results.

## G    THE USE OF LLMS

We used a large language model (LLM) to polish our initial draft and ensure grammatical correctness.

