# OpenReview forum: "Projective Symbolic Regression: Solving High-Dimensional PDE by Learning from Low-Dimensional Projections"
_ICLR.cc/2026/Conference — Submitted to ICLR 2026_

### Official Review · Reviewer_LZzP · 2025-10-27

**Soundness:** 2
**Presentation:** 2
**Contribution:** 2
**Rating:** 2
**Confidence:** 3

**Summary:**

This paper introduces PSR, an equation-discovery-based PDE solver. The core idea is to decompose the high-dimensional problem into a series of low-dimensional sub-problems by generating projections of the data. Symbolic regression is applied to each projection to find local functional components. These components are then used as new building blocks in a global symbolic regression step to compose the final, high-dimensional solution expression. The method is evaluated on a newly proposed dataset, SymPDEBench. Experiment results demonstrate that PSR outperform vanilla symbolic-regrassion-based solvers.

**Strengths:**

1. The paper is easy to follow. The figures are illustrative and helpful.
2. The idea of applying symbolic regression on projected low-dimensional spaces is conceptually novel and appealing.
3. The experiment results are strong.

**Weaknesses:**

1. The way the method picks constant values for fixed variables seems not well-defined. For continuous variables, the concept of a "most populated value" is not meaningful, as individual measurements are often unique. Furthermore, this approach is incompatible with irregularly sampled data or non-grid structures, as it implicitly assumes the availability of a complete, structured data slice when a variable is held constant.
2. Cases are that many PDEs don't have closed-form expressions for the solutions. In this cases, the assumption of sparsity doesn't necessarily hold. Furthermore, the perfomance of equation discovery on high-dimensional data isn't satisfying enough. This further limits PSR's application.
3. The training pipeline isn't clear. The authors present distinct losses for local component discovery and global expression composition but do not clarify whether these are optimized simultaneously or sequentially. Additionally, the paper fails to detail the sampling strategy for projected data—a critical omission, as projection is central to the method. The authors should provide theoretical justification or proof that the set of projections used is sufficient to guarantee the recoverability of the original PDEs.

**Questions:**

1. How do you expect the performance of PSR on PDEs without closed-form solutions?
2. Are there any efficiency improvement/degradation compared to the baselines?
3. How PSR performs from PDE discovery's perspective? What is the recall, precision and accuracy?

---

### Official Review · Reviewer_aU7S · 2025-10-27

**Soundness:** 1
**Presentation:** 3
**Contribution:** 1
**Rating:** 2
**Confidence:** 4

**Summary:**

The paper proposes Projective Symbolic Regression (PSR), which is a hierarchical method for discovering symbolic solutions to high-dimensional PDEs. The methods works by, first, projecting the high-dimensional data to lower dimensional subspaces, then, applying a local symbolic regression approach to discover the functional components in each projection, and finally, discover a composition function of these components through a global symbolic regression approach constrained via a PDE residual loss. The approach is evaluated on a new benchmark, called SymPDEBench, that contains five PDE types in different dimensions.

**Strengths:**

The paper contains some novel ideas, such as

- the MSEDI Loss which is new to symbolic regression and a clever solution to encourage dynamically correct solutions in local SR.

- the specific two-stage hierarchy approach which addresses the important question of how to choose the form of the PDE solution, as well as, the form of each component of the solution. This was initially explored in the AI Feynman paper, but adapted to PDEs.

- the Projection Bases decomposition is interesting, however in my understanding not that novel as HD-TLGP does something similar.

The paper is very clearly written, with nice graphics, but some parts are confusing in my opinion.

**Weaknesses:**

In my understanding there are the following major issues with the paper:

- There seems to be a confusion between Symbolic Regression and Symbolic PDE solution. Symbolic regression can be naively thought as a map from a field or generally some numerical data to a symbolic expression, so the model discovers an expression that when evaluated, it fits the given numerical data (the so-called inverse problem). On the other hand, symbolic solution of PDEs objective is to find an expression that satisfies a given PDE for given initial/boundary conditions and a given domain (the so-called forward problem). If one has already access to the numerical solution, why even run this symbolic process? You get an approximation of an approximation. Moreover, it is not clear to me how f_1, f_2, and f_3 are defined in Table 9 in Appendix C to verify if the solution is indeed more interpretable.

- The PDEs considered are trivial, uncoupled and separable in dimensions. This makes many claims in the paper to be inaccurate. For example,
              - the method mitigates the curse of dimensionality: only for pre-separable cases of PDE solutions
             - Physics principles enable decompositions: True by construction in the provided examples, unjustified for other cases.
             - Comprehensive PDE test suite: Only separable examples of simple solutions.
             - Physical Insight : Unjustified, because the f’s are not defined (to the best of my knowledge. Maybe I missed it.)

- The dictionary terms in Table 5 are tailored for each PDE and chosen contain exactly the operations present in the specific PDE solution. This is a major issue because it invalidates the claim of reducing the complexity of the space as the space spanned by the basis does not contain redundant operations.

- The sparsity of effect, decomposing complex effects into simpler ones, might make sense in the operator level. For example, you can have a diffusion equation u_t - D u_xx = 0 and then add a reaction term f(u) and have a diffusion-reaction u_t - D u_xx - f(u) = 0, which is exactly this decomposing/composing approach. However, that does not translate to the solution. For example, if f(u) = r u (1-u) (a Fisher-KPP equation) has a traveling wave solution u(x, t) = 1/(1 + exp(\lambda (x - ct)) ) (sorry for the lack of rigor but I do not want someone to miss the point) which is not a composition of a diffusion and a reaction solution and not separable. So, the core of the approach only works for a very small subset of PDEs that are either additive or multiplicative separable.

-  The sensitivity analysis is misused. The Sobol indices quantify “how much varying x_i affects the output when the other inputs  are random” it is not connected with the separability of the solution. Let me give you an example to explain this. Separability means that higher order interactions between inputs are exactly zero, while sparsity of effect (as considered here through Sobol indices) means that higher order interactions are small. Consider u(x,y) = 50x + 50y + 0.5 xy. It is obvious that almost all the variance comes from the main effects, however this solution is not separable.

And minor:

- The first three paragraphs of the introduction present a contradiction. It is not clear is the authors aim to find a more efficient approximate method or analytical solutions.

- The partial differential equation problem is not properly defined in Section 2.1. The authors need to define boundary/initial conditions and the domain to define a well-posed problem, else it is not meaningful.

- Figure 1 has notational mismatch between variables.

- The claim that the projection approach can be applied to any SR method is not true. It cannot be applied to methods that consider pre-training, see Lampe and Charlton. So, the authors need to remove or adjust the claim.

**Questions:**

I have the following questions:

- How are the dictionary of operators chosen for each PDE because it seems that it is case dependent. What happens if you use the complete universal dictionary for all PDEs?

- The process starts consider that the analytical (or good approximate solution) is known. What happens if it is not? What happens if you do not consider the numerical solution at all?

- What is the wall-clock time required for PSR (all stages)?

- How is the projection strategy selected? It does not seem like there is a formal way to do this.

- Why are there huge jumps in accuracy PSR for different examples and methods, e.g. Wave 2-2D PSR: 1e+04, NetGP: 9.09e-16.

---

### Official Review · Reviewer_eG1i · 2025-10-31

**Soundness:** 3
**Presentation:** 3
**Contribution:** 2
**Rating:** 4
**Confidence:** 4

**Summary:**

This paper introduces Projective Symbolic Regression (PSR), a novel framework for discovering interpretable, symbolic solutions to high-dimensional partial differential equations (PDEs). The core problem addressed is the curse of dimensionality, which makes traditional symbolic regression (SR) intractable for such problems. PSR tackles this by proposing a hierarchical, "divide-and-conquer" strategy. First, it projects the high-dimensional solution data onto several low-dimensional subspaces. Then, a local SR engine discovers functional components within each projection. Finally, a global SR engine composes these components into a unified, high-dimensional expression, guided by minimizing the PDE residual to ensure physical validity. The authors demonstrate on a newly proposed benchmark, SymPDEBench, that PSR outperforms existing methods in terms of predictive accuracy and provides interpretable models that reveal the compositional structure of the underlying physics.

**Strengths:**

- Novel and Well-Motivated Framework: The proposed PSR framework is an elegant and novel approach to a challenging problem. The core idea of decomposing a high-dimensional search space into low-dimensional projections is well-motivated by the "sparsity-of-effects" principle, which is a sound assumption for many physical systems. This makes the decomposition strategy more tailored and effective for PDEs compared to generic divide-and-conquer approaches.

- Addresses a Key Limitation of Prior Work: A major strength is that PSR does not rely on known low-dimensional analytical solutions, a significant limitation of prior specialized methods like HD-TLGP. This makes PSR far more practical and applicable to real-world scientific problems where such prior knowledge is often unavailable.

- Clarity: The paper is well-written and clearly structured. The methodology is explained with helpful illustrations (Figure 1, 3).

**Weaknesses:**

- Ambiguity in Projection Strategy Selection: The success of PSR seems to critically depend on the choice of "Projection Strategies" (S_j). The paper states these are "predefined" and gives an example of projecting onto (spatial_dim, time) subspaces. However, it is not clear how these strategies are determined in a general case. This raises questions about the level of manual effort or domain expertise required. The framework's performance on problems with strongly coupled variables that are not easily separable into the chosen low-dimensional projections (e.g., solutions involving terms like sin(x1*x2)) is not discussed.

- Scalability to Very High Dimensions: The paper frames the problem as "high-dimensional," with experiments conducted on 2D and 3D spatial problems. While this is a clear advance for the field of SR for PDEs, the combinatorial growth of possible low-dimensional projections (e.g., all pairs of variables) could pose a challenge for scaling to much higher dimensions (e.g., > 5). A discussion on the computational complexity and practical limits of scalability would strengthen the paper.

- Dependence on the Sparsity-of-Effects Assumption: The method's foundation is the sparsity-of-effects principle. The paper would benefit from a discussion of its failure modes. How would PSR perform on PDEs whose solutions are dominated by high-order, non-separable interactions where this assumption is violated? Presenting such a case, even if performance is degraded, would help to better define the scope and limitations of the approach.

**Questions:**

- Could the authors elaborate on the process of defining the "Projection Strategies"? Is this currently a manual process requiring expert knowledge for each new type of PDE, or is there a heuristic or automated way to generate them? How would the method cope if an inappropriate set of projections is chosen?

- Regarding the local discovery stage, the use of MSEDI as a proxy for derivative constraints is clever. The paper states that the "full high-dimensional PDE residual cannot be computed on a low-dimensional slice". Could you explicitly confirm if this is because the required partial derivatives (e.g., ∂u/∂x2 on a slice where x2 is fixed) are undefined or zero, making the PDE operator ill-posed on the slice?

- The results in Table 3 are very impressive for revealing the compositional structure. However, in some cases, the discovered expression seems more complex than a simple addition. Does this reflect redundancy in the discovered local components f_i, or is there a deeper reason? Does the framework include mechanisms to simplify the final composed expression?

- Following up on the scalability weakness, what is the computational cost of PSR compared to the baselines, especially as the number of spatial dimensions increases from 1 to 3? An analysis of how the runtime of each stage (projection, local SR, global SR) scales with dimensionality would be insightful.

---

### Official Review · Reviewer_G18x · 2025-11-01

**Soundness:** 2
**Presentation:** 2
**Contribution:** 1
**Rating:** 2
**Confidence:** 4

**Summary:**

This work proposes to use symbolic regression to learn solutions to PDEs by first fitting subsets of the variables. The authors test on 1D, 2D, and 3D linear PDEs.

**Strengths:**

The authors perform a large number of numerical experiments to test their approach.

**Weaknesses:**

Conceptually, it is not clear why performing symbolic regression for PDE solutions is useful. For most real-world applications, the solutions to PDEs are not expected to be analytic.

The authors claim to be dealing with high-dimensional PDEs, but only go up to 3D PDEs. This means there are at most 3 spatial variables and one time, which is not at all a high-dimensional problem that would require methods like the proposed projection method in order to be feasible.

Formally speaking, "projection" is not even the right word for what seems to be happening. The authors simply use a subset of the data that has certain variables fixed. This drastically reduces the size of the dataset, throwing away almost all of the data (especially in high dimensions), and seems very inefficient. This approach also cannot handle irregularly sampled data, which is often the case for very high-dimensional PDEs.

Furthermore, the authors only test on linear PDEs (including the reaction-diffusion PDEs, which only use linear reaction terms), which have general solutions that are very well understood. There is no study of nonlinear PDEs with complex dynamics that have potentially non-trivial analytic solutions.

**Questions:**

1. What is the use case that you see for this kind of method?
2. Does symbolic regression still work without the proposed projection method (directly using global symbolic regression) on your test cases?

---

### Meta-Review · Area_Chair_9jgC · 2026-01-02

**Summary:**

Here is a summary of the main massive concerns of the reviewers:
* The general set-up was questioned, since it doesn't make sense to introduce a symbolic regression approach to solutions of PDEs (see the citation below).
* Even the word "projection" raised concerns, since what the authors do is not really a projection.
* Applicability in very high dimensions
* Only trivial PDEs were considered

**Reviewer Concerns:**

The concerns of the reviewers were massive, and the authors did not address any of those.

The following sentence of Reviewer G18x sums it up very nicely: "Conceptually, it is not clear why performing symbolic regression for PDE solutions is useful. For most real-world applications, the solutions to PDEs are not expected to be analytic." Also to me the paper does not make sense.

**Reviewer Scores:**

It’s really hard to say how any reviewer would have changed their score if they had taken part more fully in the discussion. Without hearing it from them directly, anything we write here would just be guesswork.

For the present paper, the scores were 2,4,2,2. Since no concerns were addressed, the scores presumably remain, yielding a clear reject.

---

### Decision · Program_Chairs · 2026-01-26

Reject